# Long-term neurological complications in COVID-19 survivors: study protocol of a prospective cohort study (NeurodegCoV-19)

Natália Araújo ,[1,2,3] Isa Silva,[1,2] Patrícia Campos,[1,2] Rita Correia ,[1,2]
Margarida Calejo,[4] Pedro Freitas,[5] Mariana Seco,[4] Luís Ribeiro,[4]
Ana Rute Costa,[1,2,3] Samantha Morais,[1,2] Susana Pereira,[1,2,6]
João Firmino-Machado ,[1,2,7] Rita Rodrigues,[8] Joana Pais,[1,2] Luís Ruano,[1,2,8]
Nuno Lunet ,[1,2,3] Vítor Tedim-Cruz[1,2,4]

For numbered affiliations see end of article.

**Correspondence to**
Dr Natália Araújo;
natalia.araujo@ispup.up.pt

## ABSTRACT

**Background** Evidence suggests an association between SARS-CoV-2 infection and worse performance on cognitive tests, and a higher risk of Parkinson's disease (PD) and dementia up to 6 and 12 months after infection, respectively. Longer follow-ups with comparison groups are needed to clarify the potentially increased risk of neurodegenerative diseases in COVID-19 survivors, namely those infected before mass vaccination.

**Methods** A prospective study started in July 2022 with four cohorts of 150 individuals each, defined according to SARS-CoV-2 infection and hospitalisation status between March 2020 and February 2021: cohort 1—hospitalised due to SARS-CoV-2 infection; cohort 2—hospitalised, COVID-19-free; cohort 3—infected, not hospitalised; cohort 4—not infected, not hospitalised. Cohort 2 will be matched to cohort 1 according to age, sex, level of hospitalisation care and length of stay; cohort 4 will be age-matched and sex-matched to cohort 3. Baseline, 1-year and 2-year follow-up evaluations will include: cognitive performance assessed with the Montreal Cognitive Assessment (MoCA) and neuropsychological tests; the assessment of prodromal markers of PD with Rapid Eye Movement Sleep Behaviour Disorder single-question Screen and self-reported olfactory and gustative alterations; screening of PD with the 9-item PD screening questionnaire; gait evaluation with Timed Up&Go test. Suspected cases of cognitive impairment and PD will undergo a clinical evaluation by a neurologist. Frequency measures of neurological complications, prodromal markers and diagnoses of dementia and PD, will be presented. The occurrence of cognitive decline—the difference between baseline and 1-year MoCA scores 1.5 SD below the mean of the distribution of the variation—will be compared between cohorts 1 and 2, and cohorts 3 and 4 with OR estimated using multivariate logistic regression.

**Ethics and dissemination** This study received ethics approval from the Ethics Committees of the health units Unidade Local de Saúde de Matosinhos and Centro Hospitalar de Entre Douro e Vouga, and informed consent is signed for participating. Results will be disseminated among the scientific community and the public.

## STRENGTHS AND LIMITATIONS OF THIS STUDY

⇒ Prodromal markers of Parkinson's disease and dementia, as well as diagnoses of these diseases, will be evaluated with screening tests, neuropsychological tests and a clinical evaluation by a neurologist.

⇒ In addition to assessing episodic cognitive impairment 2–3 years after SARS-CoV-2 infection, we will evaluate cognitive performance over 2 years and compare individuals who had mild/severe SARS-CoV-2 infections with the general population or a group of hospitalised patients.

⇒ Data on cognitive performance will only be obtained after SARS-CoV-2 infection, limiting the assessment of causality between SARS-CoV-2 infection and cognitive impairment.

⇒ Sample size, calculated to detect an association between SARS-CoV-2 infection and cognitive decline defined based on the variation in Montreal Cognitive Assessment score over 1 year, may not be large enough to identify a significant association between COVID-19 and dementia or Parkinson's disease, requiring a longer follow-up and/or external collaborations.

⇒ Misclassification of the exposure in the comparison groups and the possibility of infection after vaccination may lead to a possibly weaker association than if comparison groups had been completely COVID-19 free over all the study period.

## INTRODUCTION

COVID-19 mortality is still high in some countries and the response to the pandemic is not uniform around the world. Mass vaccination, booster vaccine doses offered to the most vulnerable population subgroups, access to adequate treatment of COVID-19, and less aggressive variants have allowed many countries to face new cases of SARS-CoV-2 infection with a lower load on the health system.[1] However, the long-term health consequences

BMJ

of SARS-CoV-2 or long COVID may represent an important health problem. Long COVID is defined by the WHO as the continuation or development of new symptoms 3 months after the initial SARS-CoV-2 infection, with these symptoms lasting for at least 2 months with no other explanation.[2] It may affect 45% of hospitalised and non-hospitalised SARS-CoV-2 infected individuals within 4 months postinfection,[3] and its worldwide prevalence at 120 days from infection, was estimated at 0.49 (higher prevalence among hospitalised patients: 0.54 vs 0.34), accounting for nearly 38 million cases.[4] Fatigue and cognitive complaints are the most frequent symptoms of long COVID[4] and were associated with difficulties in returning to full-time employment.[5]

Although COVID-19 is a respiratory disease, neurological manifestations during the acute phase of SARS-CoV-2 infection are common.[6] A few cases of parkinsonism developing between 5 and 32 days after initial symptoms of SARS-CoV-2 infection were described, suggesting parkinsonism as a consequence of mild and severe COVID-19 infection.[7–11] In a large retrospective study (January–December 2020), COVID-19 was also associated with an increased risk of neurological conditions during the 6 months following infection, namely, parkinsonism (increased risk in 45%) and dementia (increased risk in 71%), when compared with matched controls of individuals infected with other respiratory tract infections, although the incidence of parkinsonism and dementia were low: 0.11% and 0.67%, respectively, at 6 months after infection with SARS-CoV-2.[12] A four-point decrease in the Montreal Cognitive Assessment (MoCA) was observed in 21% and 2% of seropositive symptomatic and seronegative asymptomatic individuals for SARS-CoV-2 infection, respectively, from a prepandemic to a postpandemic evaluation (253.4 person-years of follow-up),[13] and a cumulative incidence of cognitive impairment (assessed with a telephone interview for cognitive status) of 12.45% was reported 12 months after hospital discharge in a large study in Wuhan, China.[14] Neurological symptoms after infection have been corroborated with brain microstructural changes on imaging exams, namely in the central olfactory system, 3 months after recovery from COVID-19, with no differences according to COVID-19 severity.[15] Indeed, there is evidence from in vivo (humans), in vitro and animal studies, that the novel COVID-19 has the potential to be neuroinvasive.[16]

Early detection of neurodegenerative complications during prodromal phases of disease is essential for implementing disease-modifying interventions. Impaired olfaction and rapid eye movement (REM) sleep behaviour disorder (RBD) are prodromal markers with the highest sensitivity and specificity, respectively, to predict Parkinson's disease (PD).[17 18] Non-dementia cognitive impairment and low gait speed can be used to predict dementia.[19–21] Moreover, cognitive decline over 1 year, that is, a deterioration in cognitive performance over this period, may help in identifying individuals who have normal cognitive performance but who will develop a long-term declining trajectory of cognitive performance.[22] Therefore, this study protocol describes the ongoing prospective cohort study NeurodegCoV-19 which aims to:

1. Quantify the association between SARS-CoV-2 infection and 1-year cognitive decline, comparing individuals infected with SARS-CoV-2 virus (hospitalised patients or cases with clinical follow-up in the community) and a control group (hospitalised patients or the general population).
2. Describe trajectories of cognitive performance over 2 years.
3. Estimate frequency measures of the occurrence and persistence of long-term neurological symptoms and diagnoses, and prodromal markers of PD, in COVID-19 survivors.
4. Explore the associations between SARS-CoV-2 infection and prodromal markers of PD.

In this study, the reference period to define SARS-CoV-2 infection/COVID-19 hospitalisation was the first year of the pandemic. Infections in this period are more likely to be associated with an increased risk of neurological complications because vaccines were essentially unavailable and because in vitro and animal models have shown the lower neurotropism and neurovirulence of the Delta and Omicron BA.1 variants.[23] Recruitment began on 1 July 2022 and is expected to end in July 2023. Therefore, the baseline evaluation is being performed 2–3 years after SARS-CoV-2 infection, and the 2-year follow-up is expected to be complete in July 2025.

## METHODS AND ANALYSIS
### Setting
This prospective cohort study is based on the population of Matosinhos, one of the counties of the Porto Metropolitan Area, Northern Littoral of Portugal, the region with the highest number of COVID-19 cases during the first months of the pandemic (326 378 cases until 28 February 2021).[24] Nearly all dwellers in Portugal are enrolled in public primary healthcare units, either as regular users, sporadic users or whether they have never used this service, and receive a unique identification number in the National Health System. Those enrolled at these units of Matosinhos (176 299 inhabitants enrolled in 2020, and 177 715 in 2021) were the source population for this study. The local health unit of Matosinhos, Unidade Local de Saúde de Matosinhos (ULSM), integrates primary and continuous palliative care units and its hospital, Hospital Pedro Hispano, under a single entity and the same management. The clinical pathology laboratory of ULSM performed all COVID-19 diagnostic tests prescribed by family doctors and public health services of the ULSM, except when results could not be obtained in 24 hours due to an overload of testing; private laboratories collaborated in the latter cases.

Additionally, the hospital unit Centro Hospitalar de Entre Douro e Vouga (CHEDV) in Santa Maria da Feira,

another county of the Porto Metropolitan Area, which includes three hospitals—Hospital de São Sebastião, E.P.E. (in Santa Maria da Feira), Hospital Distrital de São João da Madeira, and Hospital São Miguel (in Oliveira de Azeméis)—also participated for the selection of hospitalised participants. CHEDV does not include primary healthcare nor continuous palliative care units.

## Study design

This is a prospective study with a baseline assessment 2–3 years after SARS-CoV-2 infection and follow-up evaluations 1 and 2 years later. Baseline assessments started on 1 July 2022, and are expected to end on July 2023. Participants were retrospectively identified from administrative lists of ULSM and CHEDV, and according to the exposure definition, as follows: having a first diagnosis of SARS-CoV-2 infection (detection of RNA virus in nasopharyngeal or oropharyngeal swabs, with Real-Time PCR), before any vaccination, during the period from March 2020 (first case diagnosed in Portugal on 2 March) to February 2021 (complete vaccination uptake reached 3% and partial vaccination, 6%, on 28 February 2021).[25] Exposed individuals are grouped according to hospitalisation at Hospital Pedro Hispano or at one of the three hospitals of CHEDV due to infection with COVID-19— cohort 1, or clinical follow-up in the community—cohort 3. The unexposed group is defined as having no diagnosis of SARS-CoV-2 infection in the same period, nor after if unvaccinated, and with at least the primary scheme of vaccination, to avoid the inclusion of controls who could get infected during the follow-up period without being vaccinated. Unexposed individuals were also grouped according to having been hospitalised at Hospital Pedro Hispano or at CHEDV during the same period—cohort 2, or not —cohort 4.

According to instructions from the Ministry of Health, from 27 October 2020, all patients had to be tested for SARS-CoV-2 infection before hospital admission.[26] Therefore, efforts were initially made to select participants infected and/or hospitalised after this date to ensure the non-infection status of the hospitalised comparison group. However, due to a low number of eligible participants for cohort 1, we extended the period to March 2020.

Misclassification regarding the non-exposed comparison groups cannot be excluded as the seronegativity for SARS-CoV-2 infection of the comparison groups could not be confirmed due to the retrospective identification of the participants. Although the National Serological Survey reported a low seroprevalence (2.9%) in May–June 2020,[27] it was higher in February–March 2021 (13.5% due to natural infection).[28]

## Participants

Eligible participants are 18 or older at the COVID-19 diagnosis (similar age at baseline for the comparison groups). General and cohort-specific eligibility criteria are checked by consultation of medical records and complemented with information from the participant during recruitment.

## SARS-CoV-2-infected patients

Participants with a positive test for SARS-CoV-2 infection between March 2020 and February 2021 were retrospectively identified from the ULSM and CHEDV databases, which, in the case of ULSM, includes tests sampled at the ULSM (96.7%), at temporary emergency units outside the hospital (0.3%,) and the drive-in community centre (3.1%), whereas all tests from CHEDV were sampled at the hospitals.

Individuals infected with SARS-CoV-2 were included into one of two groups: cohort 1: individuals hospitalised at Hospital Pedro Hispano or at CHEDV due to COVID-19 infection, and cohort 3, individuals with clinical follow-up of COVID-19 in the community (asymptomatic or not). Individuals for cohort 1 were identified in two steps: (1) exclusion, from the list of hospitalisation episodes at Hospital Pedro Hispano or at CHEDV, of those with a description of Homogeneous Diagnostic Groups surely not compatible with virus infection (eg, childbirth); (2) consultation of medical records of hospitalisation episodes not excluded in the previous step, to identify if hospital admission was due to COVID-19, the highest level of care (general ward, intermediate care, intensive care unit) and length of stay (figure 1). All individuals identified were contacted for recruitment.

Individuals for cohort 3 were those on the ULSM list of positive tests for SARS-CoV-2 infection and not on the ULSM list of hospitalised patients (criterion confirmed by the participant) (figure 1). Three pools of individuals were selected from the pandemic's first, second and third waves (earliest 30 diagnosis dates in 2020; all cases from 27 October 2020 to 3 November 2020, and the latest 30 diagnosis dates in February 2021).

## Unexposed comparison groups

A comparison group of hospitalised patients (cohort 2), with no positive test for SARS-CoV-2 infection between March 2020 and February 2021, never tested positive afterwards or infected after complete primary vaccination, was identified from the list of hospitalised patients between March 2020 and February 2021 who were not on the list of positive tests for COVID-19 for this period (figure 1). Individuals for cohort 2 will be matched with individuals from cohort 1 regarding the hospital (Hospital Pedro Hispano or CHEDV), age (18–24, 25–34, 35–44, 45–54, 55–64, 65–74 and ≥75 years), sex, level of hospitalisation care (general ward, intermediate care, ICU), number of days of hospitalisation (1, 2–7, 8–15, 16–30 and >30 days), and month of hospital admission.

For comparison with cohort 3, cohort 4 will include (baseline) age-matched and sex-matched individuals randomly selected from the pool of individuals enrolled at ULSM, who were not on the list of a positive diagnostic test for COVID-19 between March 2020 and February

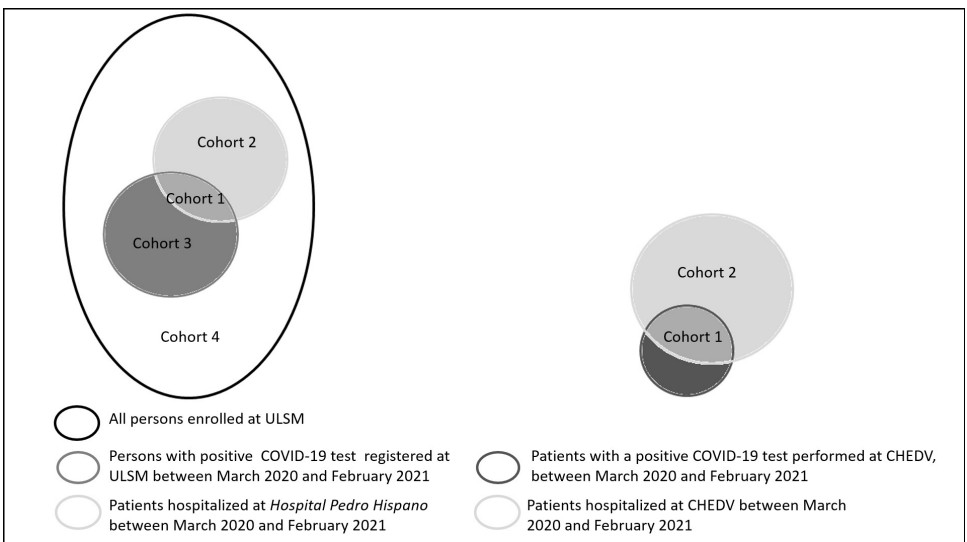

**Figure 1** Identification of potential participants from the lists of COVID-19 test results and hospitalisation episodes during the period from March 2020 to February 2021, and from the list of individuals enrolled at primary healthcare units of Unidade Local de Saúde de Matosinhos (ULSM). Cohort 1 includes individuals with a positive diagnosis test for COVID-19 registered at ULSM and hospitalised at hospital Pedro Hispano and those with a positive test performed at Centro Hospitalar de Entre Douro e Vouga (CHEDV) and hospitalised at one of the three hospitals of CHEDV. Cohort 2 includes individuals with no positive diagnosis test for COVID-19 and hospitalised at hospital Pedro Hispano and at CHEDV. Cohort 3 includes individuals with a positive diagnosis test for COVID-19 registered at ULSM and with clinical follow-up in the community. Cohort 4 includes individuals enrolled at ULSM with no positive diagnosis test for COVID-19 and without any hospitalisation episode between March 2020 and February 2021. Areas are not proportional to the number of cases.

2021, nor on the list of hospitalised patients in the same period (figure 1).

### Exclusion criteria

We will exclude: (1) individuals diagnosed with a condition impairing cognitive function or enrolled in treatments for cognitive decline (consumption of Memantine, Rivastigmine or Donepezil; participation in psychotherapeutic interventions for clinical or research purposes), or under clinical neuropsychological assessment, or with a history of substance abuse (registered in the medical file or spontaneously reported by the participant); (2) individuals with less than 1 year of formal education, and primary education (grades 1–4) not in the Portuguese education system (criteria for the use of cognitive tests) and (3) individuals with a motor, visual or auditory problem incompatible with the performance of the cognitive tests.

Individuals living at senior homes are excluded, as well as those infected between the two doses of the primary vaccination scheme or within the 14 days following vaccination.

The report from the participant of an infection with SARS-CoV-2 virus during the period from March 2020 to February 2021 that could not be confirmed with a registry of a positive result at the RT-PCR test, or a hospitalisation episode in a private hospital reported by the participant during this period, will lead to the exclusion of the participant.

### Recruitment

Phone calls to contact the participants are performed during the day (morning, afternoon and evening) and with repeated attempts, in case of failure. A trained member of the research team conducts a structured conversation aiming to confirm the eligibility of participants, explain the objectives and procedures of the study, and schedule the in-person evaluation at Hospital Pedro Hispano or Hospital São Sebastião. Reasons for not participating (impossible to contact, exclusion criteria or refusals) are registered for all potential participants identified for the cohorts 1–4.

### Baseline and follow-up evaluations

Participants are evaluated in person, at Hospital Pedro Hispano or Hospital São Sebastião, at baseline, that is, approximately 2–3 years after the index date, which corresponds to the date of the first positive test for SARS-CoV-2 infection, for cohort 3, the date of hospital admission due to acute COVID-19, for cohort 1, and the date of hospital admission due to other reasons, for cohort 2. No index date was attributed to participants of cohort 4 as they were neither infected nor hospitalised during the period March 2020–February 2021.

Follow-up evaluations after 1 and 2 years will be similar to the baseline assessment.

### Instruments

Table 1 presents the instruments administered to all participants of the four cohorts.

**Table 1** Description of the instruments used for the evaluation of all participants

| Instrument | Description | Domains/subscales | Score |
|---|---|---|---|
| MoCA[29] | Paper and pencil cognitive screening test for evaluating mild cognitive impairment. | Attention and concentration, executive functions, memory, language, visuoconstructional skills, calculations, orientation. | Range: 0–30<br>Higher scores represent better cognitive performance. |
| 18-point CDT[52 53] | The CDT is an assessment tool where participants are asked to draw a big circle, place the clock numbers and then indicate the time as '10 past 11'. An 18-point system will be used to score the test. | Visuospatial, executive function. | Range: 0–18<br>Scoring system with three main components: (A) assessment of circle integrity (two points); (B) number placement and sequencing (six points); and (C) placement and size of the hands (six points). Additionally, there are two points for representation of the clock's centre and two points for general gestalt. |
| Brain on Track[33 34] | Computerised and self-administered test that allows longitudinal monitoring of cognitive performance. | Attention, memory, executive functions, language, calculation, constructive ability, visuospatial processing. | Range: virtually unlimited (maximum number of correct answers in a fixed time).<br>Higher scores represent better cognitive performance.<br>Scores falling below an expected performance threshold for each age/education group represent a pattern of decline in individual performance. |
| HADS[45] | Scale with 14 questions designed to assess anxiety and depression. Patients should answer considering the previous week. | Depression, anxiety. | Range (for each subscale): 0–21<br>Scores greater than or equal to 11 represent a case of anxiety or depression, as applicable. |
| PSQI[44] | Quality of sleep index with 18 questions about the sleeping habits of the patients during the previous month. | Subjective sleep quality, sleep latency, duration of sleep, habitual sleep efficiency, sleep disorders, use of medications for sleep, daytime dysfunction. | Range: 0–21<br>Scores greater than 5 indicate poor sleep quality. |
| REM Sleep Behaviour Disorder Single-Question Screen[35] | Single screening yes/no question about the classic dream-enactment behaviour in the REM behaviour disorder. | REM sleep behaviour disorder. | A positive answer suggests a presence of REM behaviour Disorder. |
| STOP-Bang questionnaire[46] | Screening tool with eight yes or no questions related to the clinical features of sleep apnoea. | Obstructive sleep apnoea (OSA). | Range: 0–8<br>Each positive answer sum 1 to the final score, with higher scores representing more risk of OSA.<br>Scores equal or greater than three indicate OSA. |
| 9-item PD screening questionnaire[38] | Screening tool with nine yes or no questions | PD | Range: 0–9<br>Higher scores indicate higher likelihood of Parkinson' Disease: unlikely if scores are 0–1 ; possible if scores are 2–4; and probable if scores are ≥5. |
| Timed "Up and Go" test[40] | Test to evaluate the ability of patients to perform sequential locomotor tasks that incorporate walking and turning. | Mobility, gait and balance (ability to stand up from the chair, walking characteristics, presence of 'freezing' walking, posture). | 5-point Likert scale (normal, discreet, mild, moderate and severe) and global gait diagnosis: normal or altered, with parkinsonism and/or osteoarticular and/or vascular and/or others features. |

CDT, Clock Drawing Test; HADS, Hospital Anxiety and Depression Scale; MoCA, Montreal Cognitive Assessment; PD, Parkinson's disease; PSQI, Pittsburgh Sleep Quality Index; REM, rapid eye movement.

## Structured questionnaire

Trained interviewers conduct face-to-face interviews using a structured questionnaire to collect baseline data on: (1) sociodemographic characteristics: age, years of education, employment status, occupation; (2) lifestyles: alcohol and tobacco consumption; (3) previous medical diagnoses and current medication intake; (4) neurological symptoms during the first infection and following recovery (cohorts 1 and 3, and individuals from the comparison groups who got infected after vaccination), and current symptoms (all participants); (5) dates of SARS-CoV-2 infection diagnoses (reinfections for cohorts 1 and 3, and infection episodes after complete vaccination, for cohorts 2 and 4 and; (6) data on vaccination uptake.

## Evaluation of cognitive performance and identification of cognitive impairment: MoCA and Brain on Track, neuropsychological evaluation and clinical evaluation by a neurologist

Cognitive performance of all participants is evaluated with the MoCA, V.7.1,[29] translated, adapted and validated in the Portuguese population.[30] MoCA is a screening

instrument for mild cognitive impairment, with good psychometric characteristics and high sensitivity.[31] It assesses visuoconstructive ability, language, attention, concentration and working memory, executive functions, delayed memory and orientation. MoCA scores range from 0 to 30, with higher values corresponding to better cognitive performance. Interviewers received training and certification from the MoCA Training and Certification Module at www.mocacognition.com, and the Manual of the Portuguese V.7.1 was discussed within the research team to clarify administration procedures and the rating process.

Two outcome measures will be used: cognitive decline and probable cognitive impairment. We will define the former based on the distribution of the MoCA scores variation over time (score at 1 year minus score at baseline) in the comparator non-exposed group (either cohort 2 or cohort 4), as being a variation in MoCA scores 1.5 SD below the mean. For defining probable cognitive impairment, we use age-specific and education-specific norms[31] and the cut-off point of 1.5 SD below the mean.

Participants with probable cognitive impairment are invited for a neuropsychological assessment, to be scheduled as soon as possible, to confirm cognitive deficits. The neuropsychological evaluation includes neurocognitive tests validated for the Portuguese population, assessing verbal and visual memory, working memory, information processing speed, executive functions and language. It also includes the assessment of subjective memory complaints, emotional functioning, functional disability, independent living skills and premorbid intelligence (table 2).

Based on the number of tests used for evaluating each cognitive domain, cut-off points (below 1 SD, 1.5 SD or 2 SD below the mean of age-norms) are applied to identify cognitive deficit[32] (table 3). Participant with cognitive deficit in at least one cognitive domain are considered to have cognitive impairment. These are referred to a clinical consultation with a neurologist of the research team at the hospital, for clinical follow-up according to the best clinical practices in use and a structured protocol of classification of the probable aetiology of the cognitive impairment. Brain imaging and laboratory tests will complement the diagnosis whenever necessary.

Additionally, cognitive performance is evaluated with the web-based remote monitoring test Brain on Track. This instrument showed good internal consistency, discriminative ability and reliability.[33 34] It assesses executive functions, short and delayed memories, attention, calculation, visuoconstructive ability and language. A complete session of tests of Brain on Track takes 24 min and includes 11 exercises with fixed duration. The participant is asked to answer to as many items as possible in each exercise, as the score is the number of correct answers. Based on this score and using a predictive model adjusted for age (to be published), a score of risk for cognitive impairment is computed. Participants with a high score of risk will be invited for the neuropsychological assessment.

Eligibility criteria to use Brain on Track are: at least 3 years of education, access to a computer at home with internet and sufficient digital literacy to log into the website, or with help from another person but with autonomy for using the mouse and to do the exercises independently.

All participants eligible to use Brain on Track are invited for remote monitoring. The first and second sessions of tests are scheduled with a 1-week interval, and the following sessions every 3 months thereafter.

## Identification of prodromal markers and screening for PD
### RBD Single-Question screen
We use the RBD Single-Question Screen[35] to identify idiopathic RBD, which is an important risk factor for PD.[36] RBD may be considered a prodromal marker of PD, dementia with Lewy bodies and multiple system atrophy, and about 75% of those suffering from RBD develop PD or a parkinsonism within about 10 years.[37] This rapid test has high sensitivity, specificity and reliability.[35]

### The nine-item PD screening questionnaire
We use the nine-item PD screening questionnaire[38] which has a sensitivity and a specificity ranging 61%–92% and 29%–92%, respectively, to identify possible cases of PD in community settings.[39] The test includes nine questions to be answered with yes or no. The sum of positive answers is the final score of the test which defines the likelihood of having PD: 0–1 (unlikely), 2–4 (possible) and ≥5 (probable).[39]

Participants screening positive for possible parkinsonism are formally evaluated by a neurologist of the research team, following usual clinical practices.

### Timed 'Up and Go' test and gait assessment
We use the Timed 'Up and Go' Test (TUG)[40] to measure the ability of participants to perform sequential locomotor tasks that incorporate walking and turning. This test was designed to measure mobility in elderly people and could be a useful tool for quantifying locomotor performance in people with PD. Patients are required to stand up from a chair, walk three metres, turn around, walk back to the chair and sit down again. One training trial is performed under instructions of a trained member of the research team and then two trials are videotaped to allow a trained neurologist of the research team to later evaluate gait and balance. The time, in seconds, necessary for the participant to perform the test is recorded. The two videotaped trials of the TUG test are evaluated using a structured report comprising: (1) four gait-related features (ability to stand up from a chair, walking characteristics, presence of 'freezing' walking, posture); (2) the rating of arms swing, asymmetry of shoulder lift, and tremor at rest on gait; (3) and a global gait diagnosis ('normal' or 'altered', with parkinsonism and/or osteoarticular and/or vascular and/or others features).

**Table 2** Description of the battery of instruments used for the neuropsychological evaluation of the participants with probable cognitive impairment*

| Instrument | Description | Cognitive domains/function | Score |
|---|---|---|---|
| SMC[54 55] | A 10-item scale regarding subjective memory complaints. | Subjective memory. | Range: 0–21 Higher scores reflect maximal memory complaints. |
| Verbal Fluency Test[56 57] | Verbal fluency tests consist of four trials, one of semantic fluency and three of phonemic fluency, of 1 min each. The participants are asked to generate the name of as many species of animals as possible in the semantic fluency test, and as many words as possible beginning with a specific letter in the phonemic fluency test, within 1 min. | Non-motor processing speed, language production and executive functions | The total trial score corresponds to the number of words correctly produced within 1 min. The total test score corresponds to the sum of the three trials. Higher scores correspond to better performance. |
| TMT[58 59] | Part A: the participants are asked to draw lines to connect 25 randomly scattered numbered circles in ascending order. They should do so as quickly as possible. Part B: the participants are asked to draw lines to connect circles in numeric and alphabetic order, alternating between numbers and letters, progressively up to no 13. They should do so as quickly as possible. | Part A: attention; visual scanning; and speed of eye-hand coordination and information processing. Part B: working memory and executive functions; particularly, the ability to switch between sets of stimuli. | Direct measures of performance: time (seconds) to complete part A and part B and performance errors during part A and part B. Derived scores: difference score (B–A), ratio score (B/A), proportion score (B–A/A), sum score (A+B), and multiplication score (A×B/100). Lower raw scores and higher adjusted scores correspond to better performance. |
| WMS-III[60 61] | Scale composed of 17 subtests that assess different memory functions using verbal and visual stimuli. Useful in the evaluation of various clinical conditions, including neurodegenerative diseases such as Alzheimer's disease. Subtests used: Logical Memory I, II; Visual reproduction I, II; Digit Span. | Verbal and visual memories, working memory. | Range: Immediate recall and delayed recall: 0–50; Auditory recognition; digit span: 0–30; Visual reproduction: 0–104 Higher scores correspond to better performance. |
| WAIS-III[62 63] | Scale composed of 14 subtests that assess the verbal component and the perceptual-motor component of intelligence in adults and older adolescents. Subtests used: Digit-Symbol-Coding; Symbol Search; Information. | Attention/concentration, executive function (sequencing), motor function; processing speed. | The number of correct symbols within the allowed time (120 s) is measured. |
| Stroop Test[64 65] | The Stroop Test evaluates the interference effects between the two cerebral hemispheres. Consist in three trials: (1) word reading; (2) colours naming and (3) identifying the colours of words, without considering the meaning of the word. | Executive functions (inhibitory control), selective attention. | Scores for each trial indicate the number of correct responses. An interference score can be generated that quantifies the participant's ability to inhibit the inappropriate response of reading the colour name as opposed to the colour of the ink used to print the colour name in the third trial. |
| Token Test-short form[66] | A test designed to assess the comprehension of commands that vary in degree of linguistic difficulty. The participant is presented with tokens of different geometric shapes, sizes and colours, and is required to perform certain acts with the tokens, such as point to tokens, touch them, pick them up, and place one token on top of another. | Attention and vigilance; verbal functions. | Range: 0–36 |

Continued

| Table 2 | Continued | | |
|---|---|---|---|
| **Instrument** | **Description** | **Cognitive domains/function** | **Score** |
| TeLPI[67] | Designed to assess premorbid intelligence, TeLPI is a Portuguese irregular word reading test. The participants are asked to read 46 irregular words, infrequent Portuguese words. | Premorbid IQ: full scale IQ, Verbal IQ, performance IQ | Range: number of errors (maximum of 46) and years of education are inserted in three linear equations to calculate the three types of IQ |
| BDI-II[68 69] | A 21-question measure assessing the presence of depressive symptoms experienced by the participant within the past week. | Emotional functioning. | Range: 0–63 A cut-off score indicative of mild depressive symptoms is greater than 10 and for severe depressive symptoms is greater than 30. |
| Barthel ADL Index[70 71] | An index to measure functional disability, focused on bodily oriented personal care. | Functional domains: feeding, incontinence, transferring, toileting, dressing, bathing. | Range: 0–100 Lower scores reflect increased disability. |
| IADL[72 73] | An eight-item scale used to assess independent living skills which include more complex activities (ie, 'instrumental activities of daily living') necessary for functioning in community settings. | Functional domains: using the telephone, shopping, food preparation, housekeeping, laundry, transport, medication, finances. | Range: 0–8 Higher scores reflect high function, independence. |

*Participants with a MoCA score below age- and education-specific values based on norms and the cut-off point of 1.5 SD below the mean.
Barthel ADL Index, Barthel Activities of Daily Living Index; BDI-II, Beck Depression Inventory – Second Edition; IADL, instrumental ADL; MoCA, Montreal Cognitive Assessment; SDMT, Symbol and Digit Modalities Test; SMC, Subjective Memory Complains scale; TeLPI, Irregular Word Reading Test; TMT, Trail Making Test; WAIS-III, Wechsler Adult Intelligence Scale – Third Edition; WMS-III, Wechsler Memory Scale – Third Edition.

## Assessment of anxiety, depression, sleep quality and obstructive sleep apnoea

Levels of anxiety and depression symptoms are measured using the Hospital Anxiety and Depression Scale (HADS)[41] validated in the Portuguese population.[42] This self-administered questionnaire includes seven items assessing anxiety symptoms and seven items for depression symptoms, resulting in the HADS-A and HADS-D subscales, respectively, rated with a four-point Likert scale (0–3). Scores of 11 or higher in the subscale are indicative of clinically significant symptoms.[42] HADS was designed to be used in patients with physical illness, and for this reason, the questionnaire does not include somatic symptoms, namely insomnia, loss of appetite and fatigue, to avoid overlapping of symptoms due to depression and

**Table 3** Criteria used for the classification of cognitive impairment, considering the number of tests administered to assess each cognitive domain

| Cognitive domain | Test | Criteria for impairment |
|---|---|---|
| Verbal memory | WMS III-Logical memory I and II | 2 scores <1.5 SD or 3 scores <1 SD |
| Visual memory | WMS III-Visual reproduction I and II | 2 scores <1.5 SD or 3 score <1 SD |
| Attention/information processing speed | WAIS III-Digit-Symbol-Coding and Symbol search | 2 scores <1.5 SD or 3 scores <1 SD |
| | Trail making test, part A | |
| | Stroop test-word reading | |
| Executive functions | Stroop test (colour naming and word colour naming) | At least three scores <1.5 SD or 2 scores <2 SD |
| | Trail making test, part B and B-A | |
| | Phonemic fluency-letters M, R and P | |
| | Phonemic fluency-categories of animals | |
| | WMS III-digit span | |
| Language | Token test-short-form | Score <2 SD |

Verbal memory and visual memory are assessed with two tasks each, and three independent scores are retrieved for each cognitive domain. WAIS III, Wechsler Adult Intelligence Scale Third Edition; WMS III, Wechsler Memory Scale Third Edition.

due to physical conditions. Several studies have shown its suitability in both clinical and community settings.[43]

To assess sleep quality, we use the self-administered questionnaire Pittsburgh Sleep Quality Index,[44] validated in the Portuguese population.[45] This instrument presents 19 items related to seven components—subjective sleep quality, sleep latency, sleep duration, habitual sleep efficiency, sleep disturbances, use of sleeping medication and day-time dysfunction. Patients are classified as poor or good sleepers (score equal or higher vs lower than five, respectively).

We use the STOP-Bang questionnaire[46] translated and validated in the Portuguese population[47 48] to screen obstructive sleep apnea. This test includes eight questions of yes or no answer. Three or more positive answers are indicative of obstructive sleep apnoea with a sensitivity of 98.6%, and eight positive answers suggest a severe condition, with a sensitivity of 80%, according to estimates obtained in a primary care setting.[48] The STOP-Bang questionnaire will also be used to control for the presence of obstructive sleep apnoea as mimic of RBD.

## Clinical information

Physicians of the research team are collecting clinical information related to the hospitalisation episode in a standardised questionnaire, including treatment (oxygen therapy, non-invasive ventilation, orotracheal intubation or invasive ventilation, vasopressors and extracardiac membranous oxygenation), pharmacological treatments for COVID-19, blood and cerebrospinal fluid test results, symptoms, modified Rankin Scale for Neurological Disability and immunosuppression status, obesity status, comorbidities and medications on admission. This information will be used to classify the hospitalisation episode according to the WHO Clinical Progression Scale[49] and to the three-risk categories of degrees of inflammation.[50]

## Data analyses and sample size
### Outcome measures
We will report on the following outcomes: neurological symptoms, prodromal markers of PD, probable cognitive impairment detected with the MoCA, cognitive decline detected with the MoCA, impairment in cognitive domains and overall cognitive impairment identified with the neuropsychological assessment, cognitive impairment and PD diagnoses by neurologist.

Frequency measures of the above-mentioned outcomes will be estimated: prevalence at baseline, and after 1 and 2 years; period prevalence over 1 and 2 years, and cumulative incidence, which will be estimated at 1 and 2 years, considering, death as a competitive event, according to the Kalbfleisch and Prentice method.[51]

Associations between SARS-CoV-2 infection and cognitive outcomes will be estimated using OR or HR with their corresponding 95% CI, obtained from multivariate logistic and Cox regressions, respectively, and considering sex, age, education, comorbidities and disease severity (not hospitalised, hospitalised in ICU, intermediate or general ward, duration of hospitalisation, WHO Clinical Progression Scale and degrees of inflammation) as covariables.

The contribution of anxiety and depression symptoms, sleep quality, and obstructive sleep apnoea to cognitive decline will also be evaluated using multivariate logistic regression.

Mixed-effect models will be adjusted to compare cognitive trajectories (considering age and education) over 2 years, according to sociodemographic and clinical characteristics, for each of the cohorts.

### Sample size calculation
In order to compare two cohorts of similar disease severity (among hospitalised—cohorts 1 and 2, or among the community—cohorts 3 and 4), considering the occurrence of cognitive decline of 7 per 100 person-years in the non-COVID-19 participants (variation in MoCA scores below 1.5 SD of the normal distribution of the variation), a ratio exposed:unexposed of 1:1, the statistical power of 80% and 5% significance level, 600 participants will be needed (150 in each cohort) to estimate the associations corresponding to an OR of at least 2.5.

This sample size will also be adequate to estimate the frequency of outcomes equal to or lower than 10% with 95% CI, and a margin of error of 7%, as well as incidence rates as low as up to 3.5 per 100 person-years with 95% CI up to 3 per 100 person-years wide.

Considering that individuals of the comparison groups as well as participants of cohorts 1 and 3, may have had SARS-CoV-2 infection episodes more than once after vaccination, including in a short time before baseline and follow-up evaluations, the associations estimated may be weaker than if the comparison groups were completely free from COVID-19. Sensitivity analyses will be performed and an increase in sample size of cohorts 2 and 4 may be needed to increase the proportion of participants without any history of COVID-19.

## PATIENT AND PUBLIC INVOLVEMENT STATEMENT
Participants were not involved in the planning of the research.

## Ethics and dissemination
This study received approval from the Ethics Committee of ULSM (117/CES/JAS), and CHEDV (39/2022), and from the ULSM Local Unit of Data Protection and Security (030/CLPSI/2021), as well as from the Data Protection Officer of the Instituto de Saúde Pública da Universidade do Porto. A Data Protection Assessment Impact was performed to evaluate and understand the processing, the need and proportionality of data processing. Therefore, this study complies with the World Medical Association Declaration of Helsinki and the General Data Protection Regulation.

Participants are informed about the institutions involved in the project and its research members; the

objectives and procedures of the research; data to be collected, namely clinical data from their electronic health record, questionnaires, cognitive tests and a video; expected individual benefits and risks (minimal); procedures to respect their right to confidentiality, their right and how to manifest this right, to abandon the study and withdraw their consent at any time and that this will not interfere with their medical care, and their right to access their data and have their data corrected and erased. After this explanation, patients clearly manifesting that they do not understand the information will not be considered eligible. Participants will have time to decide. Participation will be assumed after a participant signs the informed consent. No direct or indirect incentives for participation will be given. This is an observational study and no risk is expected. Participants with neurological complications of COVID-19 will be followed as are non-participants of the study, according to clinical practice implemented at the health institution. Therefore, participants will not be submitted to additional risk.

The findings of this project will be submitted for publication in international peer-reviewed journals, and proposed for presentation in relevant national and international conferences, which will allow for the dissemination of the main findings across the medical community. Press releases through mass media will also be issued to promote the dissemination of information relevant to the general population and policy-makers. Furthermore, the project will contribute to the training of researchers through the production of masters' dissertations and doctoral theses.

**Author affiliations**
[1]EPIUnit, Instituto de Saúde Pública, Universidade do Porto, Porto, Portugal
[2]Laboratório para a Investigação Integrativa e Translacional em Saúde Populacional (ITR), Universidade do Porto, Porto, Portugal
[3]Departamento de Ciências da Saúde Pública e Forenses, e Educação Médica, Faculdade de Medicina da Universidade do Porto, Porto, Portugal
[4]Serviço de Neurologia, Unidade Local de Saúde de Matosinhos, Senhora da Hora, Portugal
[5]Escola Superior de Saúde, Instituto Politécnico do Porto, Porto, Portugal
[6]Serviço de Neurologia, Instituto Português de Oncologia do Porto, Dr. Francisco Gentil, E.P.E, Porto, Portugal
[7]Departamento de Ciências Médicas, Universidade de Aveiro, Aveiro, Portugal
[8]Serviço de Neurologia, Centro Hospitalar de Entre o Douro e Vouga, Santa Maria da Feira, Portugal

**Contributors** NA, MC, ARC (Rute Costa), SM, SP, FM, JP, LR, NL and VT-C contributed for the conception and design of the work, as well as for obtaining funding. NA, ARC, IS and PC contributed for the design of the structured questionnaire and interview. IS and PC have recruited the participants. NA, IS, PC and RC (Rita Correia), have performed the general evaluation of participants. PC and RC have performed the neuropsychological evaluation. PF has inserted the data in the database. MS has collected clinical data of the participants. MC, RR, LR and VT-C has performed the clinical evaluation by neurologist. NA, IS, PC, RC, MC, JP, LR, NL and VT-C will analyse and interpret the data. NA, IS and PC contributed for drafting the work and all authors revised it critically for important intellectual content. All authors approved the final version of the manuscript to be published.

**Funding** This study is financed by national funds through the FCT-Foundation for Science and Technology, I.P., within the scope of the projects UIDB/04750/2020 and LA/P/0064/2020, and obtained specific FCT funds through the competitive call for Research and Development projects in all scientific domains (PTDC/SAU-EPI/6275/2020).

**Competing interests** None declared.

**Patient and public involvement** Patients and/or the public were not involved in the design, or conduct, or reporting, or dissemination plans of this research.

**Patient consent for publication** Not applicable.

**Provenance and peer review** Not commissioned; externally peer reviewed.

**ORCID iDs**
Natália Araújo http://orcid.org/0000-0002-1652-8999
Rita Correia http://orcid.org/0000-0002-3476-9040
João Firmino-Machado http://orcid.org/0000-0001-9410-633X
Nuno Lunet http://orcid.org/0000-0003-1870-1430

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
