## [Reviewer comments · BMJ Open]

ARTICLE DETAILS

TITLE (PROVISIONAL)	Long-term neurological complications in COVID-19 survivors: study protocol of a prospective cohort study (NeurodegCoV-19).
AUTHORS	Araujo, Natalia; Silva, Isa; Campos, Patrícia; Correia, Rita; Calejo, Margarida; Freitas, Pedro; Seco, Mariana; Ribeiro, Luís; Costa, Ana; Morais, Samantha; Pereira, Susana; Firmino-Machado, João; Rodrigues, Rita; Pais, Joana; Ruano, Luís; Lunet, Nuno; Tedim-Cruz, Vítor

VERSION 1 – REVIEW

REVIEWER	Muley, Arti PIMSR, Parul University, Medicine
REVIEW RETURNED	23-Mar-2023

GENERAL COMMENTS	Addition of an ethics approval and participant consent should be added.
---

REVIEWER	Ross-Russell, Amy University Hospital Southampton NHS Foundation Trust
REVIEW RETURNED	14-Apr-2023

GENERAL COMMENTS	Thank you for the opportunity to review your planned study. I will be interested to see the results when you report. I have a few comments. Firstly, I think your main limitation is that fact that infection has spread widely and included huge numbers of the general population since the beginning of the pandemic. Your comparison cohorts are allowed (I think) to have had COVID 19 since vaccination without being excluded from the control group. So your comparison is between those with COVID19 early in the pandemic, who had not received vaccination, and those who had COVID 19 after vaccination, or not at all. This will need to be clear in discussion, and should be acknowledged, unless it is possible to exclude those who have had subsequent mild infections. This may limit any significant result from being visible. Secondly, you comment on matching your hospitalised cohorts for level of care, but you do not mention their co morbidities, including neurological, or their presenting complaint/requirement for hospital care. This should ideally be matched- eg for respiratory conditions, so as to not falsely attribute complications to COVID-19. Ideally the groups should be matched as closely as possible for comorbidities, neurological conditions, risk factors for neurodegenerative conditions etc.
--

	It will be important to use standardised, accepted clinical definitions for each group. I was pleased to see careful consideration of factors that influence assessment of neurodegenerative conditions and their impact, including psychological and psychiatric assessment, and physical disability. Finally, another major limitation is the lack of a baseline assessment, pre COVID19. This will make matching cohorts, education history, risk factors etc particularly important.
--	---

REVIEWER	Hill, Rachel A. Monash University, Psychiatry
REVIEW RETURNED	15-May-2023

GENERAL COMMENTS	This is a very important study protocol that will provide novel information on the long term impacts of SARS-CoV-2 infection. minor comments: Strengths and limitations:  1. The authors list that sample size is a limitations, yet in the statistics section their power calculation suggests they are well powered to see an effect. Please clarify this stated limitation. Furthermore, this will not be mitigated by long-term follow up. I suggest alternatively that it could be mitigated by calling for further international uptake of this protocol at additional sites? 2. The last dot point in strengths and limitations should read completely COVID-19 free 3. introduction - are the authors able to report on the global prevalence of parkinsonium and cognitive decline in long-COVID ? 4. The authors report that long COVID impacts 0.5% of infected individuals - is this prevalence impacted by severity of illness? 5. To further categorize illness severity in this study the authors may wish to consider using the WHO 7 point ordinal scale for COVID-19 illness severity. 6. Please clarify for cohort 4 if participants are not hospitalized why are they enrolled at ULSM? 7. Exclusion criteria - is a history of substance abuse an exclusion criteria ? 8. Have the authors considered any brain imaging to be incorporated into the study design for those detected as showing parkinsonium or cognitive decline?
---

VERSION 1 – AUTHOR RESPONSE

REVIEWER #1 Dr. Arti Muley, PIMSR, Parul University

Comment #1

Addition of an ethics approval and participant consent should be added.

Reply to comment #1

In the Ethics and Dissemination section of the Abstract, we have added the name of the two ethics committees that approved the study, and we refer that informed consent is signed for participating. This information is also available in the Ethics and Dissemination section of the main text, namely in lines 352-358 and 367-368.

REVIEWER #2 Dr. Amy Ross-Russell, University Hospital Southampton NHS Foundation Trust
Comment #1

Thank you for the opportunity to review your planned study. I will be interested to see the results when you report.

Reply to comment #1

We thank Dr. Amy Ross-Russell for her willingness to review this manuscript. It is a pleasure to hear of her interest in our results, and we will let her know as soon as they will be reported.

Comment #2

Firstly, I think your main limitation is that fact that infection has spread widely and included huge numbers of the general population since the beginning of the pandemic. Your comparison cohorts are allowed (I think) to have had COVID 19 since vaccination without being excluded from the control group. So your comparison is between those with COVID19 early in the pandemic, who had not received vaccination, and those who had COVID 19 after vaccination, or not at all. This will need to be clear in discussion, and should be acknowledged, unless it is possible to exclude those who have had subsequent mild infections. This may limit any significant result from being visible.

Reply to comment #2

A significant proportion of the comparison groups as well as COVID-19 participants may have had SARS-CoV-2 infection episodes more than once after vaccination, including in a short time before baseline and follow-up evaluations. Despite the assumption that vaccines prevent severe and long COVID, the associations estimated may be weaker than if the comparison groups were completely free from COVID-19.

The dates of the different infection episodes and neurological symptoms during the first infection and currently are collected with the structured questionnaire applied to the participants (page 9, last paragraph, lines 189-191). Based on this information, sensitivity analyses will be performed. An increase in the sample size of cohorts 2 and 4 may be needed to increase the proportion of participants without any history of COVID-19.

We acknowledged this limitation in the last bullet of the Strength and Limitations section, and we are now adding in the Data Analysis and sample size section a last paragraph regarding this limitation and how we plan to correct it (lines 346-351). Additionally, MoCA scores obtained before the pandemic (in 2013-2015) by participants of the population-based cohort EPIPorto will allow for another comparison of the cognitive performance of individuals who survived COVID-19 two years after the infection with the general population before the pandemic but participants of EPIPorto were not re-assessed after one year and no comparison for cognitive decline (MoCA variation over one year) will be possible.

The structure of the manuscript does not allow for a Discussion section, but this will be discussed in future reports.

Comment #3

Secondly, you comment on matching your hospitalised cohorts for level of care, but you do not mention their co morbidities, including neurological, or their presenting complaint/requirement for hospital care. This should ideally be matched- eg for respiratory conditions, so as to not falsely attribute complications to COVID-19. Ideally the groups should be matched as closely as possible for comorbidities, neurological conditions, risk factors for neurodegenerative conditions etc.

Reply to comment #3

We use the matching method for more efficient recruitment of participants for the comparison groups that we randomly select from the whole pool of eligible participants. This allows us to obtain groups with adequate similarity in the most important characteristics of the participants in the groups COVID-19 hospitalized/non-COVID-19 hospitalized and COVID-19 followed in the community/non-COVID-19, non-hospitalized. These participants' characteristics are those available in the administrative lists of the hospitals (age and sex, and length of stay, and level of hospital care). However, for the

assessment of the association between SARS-CoV-2 infection and cognitive outcomes, we will adjust for possible confounding factors in the regression models, as we stated in the Data analysis and sample size section, lines 325-330. Age, education, sex, alcohol, and tobacco consumption, co-morbidities, and disease severity are considered possible confounders. Previous diagnoses of neurological/psychiatric conditions impairing cognitive function are exclusion criteria for all participants, under the exclusion criteria “individuals diagnosed with a condition impairing cognitive function” (Methods and Analysis section, Exclusion criteria, page 8, line 151-166). Other neurological conditions are registered in the list of diseases/conditions previously diagnosed.

Comment #4

It will be important to use standardised, accepted clinical definitions for each group.

Reply to comment #4

Clinical criteria are applied to identify:

1. Individuals with SARS-CoV-2 infection: a positive result at the real-time reverse transcriptase polymerase chain reaction detection of SARS-CoV-2 RNA in a nasopharyngeal or oropharyngeal swab.

2. The reason for hospitalization (due to acute infection with SARS-COV-2 or not).

These two identification requirements are applied considering the period from March 2020 to February 2021 and allow us to define the cohorts: cohort 1 is defined by a positive diagnosis test and hospitalization due to COVID-19; cohort 2 is defined by the absence of a positive diagnosis test and the existence of a hospitalization episode not due to COVID-19; cohort 3 is defined by a positive diagnosis test and no hospitalization episode; cohort 4 is defined by the absence of a positive diagnostic test and no hospitalization episode.

The information source for applying these criteria are:

- The lists of diagnostic tests for SARS-CoV-2 infection performed between March 2020 and February 2021 from the two health units ULSM and CHEDV, or the report of a positive diagnostic test in the medical file.
- The review of each medical registry associated with a hospitalization episode at ULSM/CHEDV of an eligible participant for cohort 1 or 2.

During recruitment, potential participants are asked to confirm their status regarding a positive test for SARS-CoV-2 infection in the period March 2020-February 2021 or hospitalization outside ULSM/CHEDV in the same period, as well as their vaccination status regarding COVID-19. We do not expect participants to report incorrect information on SARS-CoV-2 infection during the period of pre-massive vaccination of the population, nor on a hospitalization episode during this period or on vaccination, as for most persons, this was a memorable period. An infection during the period from March 2020 to February 2021 that could not be confirmed with a registry of a positive result at the RT-PCR test, or hospitalization in a private hospital reported by the participant during this period, will lead to the exclusion of the participant.

We have added this last exclusion criterion in lines 166-169.

We had already described the clinical criteria for the identification of COVID-19 cases, and the identification of the reason for hospitalization in lines 91-102.

Misclassification regarding the non-exposed comparison groups cannot be excluded as we have no information on the serologic status of the participants before complete vaccination. We acknowledge this limitation in lines 108-112 and in the last bullet point of the Strengths and limitations of this study.

Comment #5

I was pleased to see careful consideration of factors that influence assessment of neurodegenerative conditions and their impact, including psychological and psychiatric assessment, and physical disability.

Reply to comment #5

We thank Dr. Amy Ross-Russell for this favorable comment.

Comment #6

Finally, another major limitation is the lack of a baseline assessment, pre COVID19. This will make matching cohorts, education history, risk factors etc particularly important.

Reply to comment #6

We agree with this comment, that is reflected in the third bullet point. Indeed, comparisons of COVID-19 cases and non-COVID-19 individuals, with similar risks of cognitive decline before the pandemic (i.e, controlling for risk factors such as age, education, comorbidities), will increase the validity of our results regarding the effect of SARS-CoV-2 infection on cognitive function.

REVIEWER # Dr. Rachel A. Hill, Monash University

Comment #1

This is a very important study protocol that will provide novel information on the long term impacts of SARS-CoV-2 infection.

Reply to comment #1

We thank Dr. Rachel A. Hill for this favorable comment.

Comment #2

Strengths and limitations:

The authors list that sample size is a limitations, yet in the statistics section their power calculation suggests they are well powered to see an effect. Please clarify this stated limitation. Furthermore, this will not be mitigated by long-term follow up. I suggest alternatively that it could be mitigated by calling for further international uptake of this protocol at additional sites?

Reply to comment #2

The sample size is a limitation for the identification of a sufficient number of new dementia and Parkinson's disease diagnoses over one year of follow-up as these diseases present a low incidence rate in the population. Therefore, extending the period of follow-up will allow for an increase in the number of new cases to occur, as well as international cooperation to include new dementia/ Parkinson's disease diagnoses.

However, for sample size calculation, we chose a different outcome, cognitive decline, defined as a variation in MoCA score below 1.5 standard deviations of the mean value of the distribution of scores variation in the comparison group. This cognitive outcome may be a predictor of further cognitive impairment and we expect its occurrence to be three times more likely in the COVID-19 cases than in the comparison groups.

Additionally, to what was previously written in the Sample size section, we re-wrote the forth bullet point as follows: "Sample size, calculated to detect an association between SARS-CoV-2 infection and cognitive decline defined based on the variation in MoCA score over one year, may not be large enough to identify a significant association between COVID-19 and dementia or PD, requiring a longer follow-up and/or external collaborations."

Comment #3

The last dot point in strengths and limitations should read completely COVID-19 free

Reply to comment #3

We thank Dr. Rachel A. Hill for calling our attention to this error. We have corrected it.

Comment #4

Introduction - are the authors able to report on the global prevalence of parkinsonism and cognitive decline in long-COVID ?

Reply to comment #4

As far as we could identify studies reporting on the frequency of parkinsonism, only case reports have been published with a review summarizing 20 cases that had occurred in a few days or months following SARS-CoV-2 infection and a large retrospective study that we refer to in the introduction, reporting an incidence of 0.11% (0.08–0.14) at six months after SARS-CoV-2 infection.

In the introduction, we have added the reference to the review of the 21 cases of parkinsonism (reference 11) and the estimates of the incidence of parkinsonism and dementia, in lines 23-25. Regarding cognitive decline defined as a negative change in cognitive performance over time, we added a summary of two studies in lines 25-30. A study with longitudinal assessment of cognitive performance with the MoCA from a pre-to post-pandemic period reported a higher odd (OR=18.1, 95% CI: 1.75–188) of cognitive decline (a decrease in 4 points) in individuals who were seropositive for SARS-CoV-2 infection (mild cases of COVID-19) compared to seronegative asymptomatic individuals. Another study with individuals who had been hospitalized for COVID-19 reported a cumulative incidence of cognitive impairment of 12,45% at 12 months after hospital discharge.

Comment #5

The authors report that long COVID impacts 0.5% of infected individuals - is this prevalence impacted by severity of illness?

Reply to comment #5

The article we refer to in the manuscript reports that the pooled post COVID-19 condition prevalence was higher in hospitalized patients – 0.54% (95% CI: 0.44, 0.63) – than in non-hospitalized patients – 0.34% (95% CI: 0.25, 0.46).

We added this information in the introduction of our manuscript, lines 12-13.

Comment #6

To further categorize illness severity in this study the authors may wish to consider using the WHO 7 point ordinal scale for COVID-19 illness severity.

Reply to comment #6

We thank Dr. Rachel A. Hill for this suggestion. We are collecting several parameters of the hospitalization episode, namely those related with treatment (oxygen therapy, non-invasive ventilation, orotracheal intubation or invasive ventilation, vasopressors, and ECMO), pharmacologic treatments for COVID-19, laboratory results from blood and cerebrospinal fluid samples, symptoms, Modified Rankin Scale for Neurologic Disability and immunosuppression status, obesity status, comorbidities, and medications at admission. Therefore, we will be able to use the classification of the severity of COVID-19 episodes according to the WHO Clinical Progression Scale, as well as the three categories of degrees of inflammation.

We added this information in lines 304-313.

Comment #7

Please clarify for cohort 4 if participants are not hospitalized why are they enrolled at ULSM?

Reply to comment #7

Almost all residents in Portugal are enrolled in public primary care units and have an identification number from the National Health System that identifies them in the administrative lists of the health units. This study focuses on the population of the municipality of Matosinhos identified as people enrolled in the ULSM, whether or not they have had at least one appointment at the ULSM since they were enrolled.

In the Methods and Analysis section, Setting subsection, we added some information in the sentence in lines 68-70 "Nearly all Portugal dwellers are enrolled in public primary healthcare units, whether as regular users, sporadic or have never used this service, and receive a unique identification number in the National Health System." to help clarify this point.

Comment #8

Exclusion criteria - is a history of substance abuse an exclusion criteria?

Reply to comment #8

We have updated the exclusion criteria in lines 159-160 to include substance abuse as documented information in the medical file or as spontaneously reported by patients at recruitment after our explanation of the study and the need to collect information on factors that may affect cognition. We do not ask about substance abuse.

Comment #9

Have the authors considered any brain imaging to be incorporated into the study design for those detected as showing parkinsonism or cognitive decline?

Reply to comment #9

Patients with cognitive impairment in at least one cognitive domain or with parkinsonism will be invited for a clinical consultation by a neurologist who will prescribe imaging exams and laboratory measurements according to usual clinical practice.

We added this information in line 231.